# Human Interaction Recognition Based on Whole-Individual Detection

**DOI:** 10.3390/s20082346

**Published:** 2020-04-20

**Authors:** Qing Ye, Haoxin Zhong, Chang Qu, Yongmei Zhang

**Affiliations:** School of Information Science and Technology, North China University of Technology, Beijing 100144, China; frank097u7@163.com (H.Z.); qc_qearl@163.com (C.Q.); zhangym@ncut.edu.cn (Y.Z.)

**Keywords:** human interaction recognition, whole-individual detection, parallel multi-feature fusion network, Gaussian model downsampling

## Abstract

Human interaction recognition technology is a hot topic in the field of computer vision, and its application prospects are very extensive. At present, there are many difficulties in human interaction recognition such as the spatial complexity of human interaction, the differences in action characteristics at different time periods, and the complexity of interactive action features. The existence of these problems restricts the improvement of recognition accuracy. To investigate the differences in the action characteristics at different time periods, we propose an improved fusion time-phase feature of the Gaussian model to obtain video keyframes and remove the influence of a large amount of redundant information. Regarding the complexity of interactive action features, we propose a multi-feature fusion network algorithm based on parallel Inception and ResNet. This multi-feature fusion network not only reduces the network parameter quantity, but also improves the network performance; it alleviates the network degradation caused by the increase in network depth and obtains higher classification accuracy. For the spatial complexity of human interaction, we combined the whole video features with the individual video features, making full use of the feature information of the interactive video. A human interaction recognition algorithm based on whole–individual detection is proposed, where the whole video contains the global features of both sides of action, and the individual video contains the individual detail features of a single person. Making full use of the feature information of the whole video and individual videos is the main contribution of this paper to the field of human interaction recognition and the experimental results in the UT dataset (UT–interaction dataset) showed that the accuracy of this method was 91.7%.

## 1. Introduction

Human motion recognition is a key technology for intelligent video surveillance. It is widely used in various scenarios such as human–computer interaction [1], motion analysis [2,3,4,5,6,7], intelligent monitoring, gesture recognition [8,9], and facial emotion recognition [10,11,12,13]. Human motion recognition is divided into single-person motion recognition and multi-person interactive motion recognition. At present, the research on human interaction recognition has gradually attracted the attention of researchers, and has obtained certain research results [14,15,16,17,18,19,20]. The algorithm framework of human interaction recognition is generally divided into whole human interaction recognition and individual human interaction recognition. This paper combined the characteristics of the two ways, and proposed a method of human interaction recognition based on whole and individual detection.

### 1.1. Related Work

At present, there are many practical algorithms and networks that can be used for human motion video recognition research. Zhao et al. [21] believe that convolution directly in the time dimension implies a strong assumption that the features between frames are well aligned, but in fact, the person or the object may perform large displacement or deformation in the video. They proposed considering trajectory convolution as a special case of 3D deformable convolution, which provides the offset amount by time series information, so that the trajectory convolution can be easily realized based on the code of the deformable convolution. Silambarasi et al. [22] proposed a video representation method of 3D volume space and human motion trajectory that projected the video onto three different views, called the 3D space–time plane, and used time-view motion tracking to identify various human behaviors. Du Tran et al. [23] proposed a new form of convolution to deal with spatiotemporal information and used 3D convolution to process spatio-temporal information and made further improvements in the form of 3D convolution. They proposed an improved model: 2D + 3D, called mixed convolution, which has the benefit of reducing parameters and maintaining performance. Chen et al. [24] made the convergence speed as fast as possible and proposed a multi-fiber network. This splits the complex network into a lightweight network and uses the information flow between the fibers to introduce the multiplexer module. Wang et al. [25] proposed a non-local operation as a generic family of building blocks for capturing long-range dependencies. This was then a concrete introduction to their method, which was inspired by the classical non-local means method of computer vision.

In the field of human interaction recognition, there are many practical algorithms and networks that can be used in human interactive video recognition research. Nilar Phyo et al. [26] applied deep learning technology over the skeleton motion history image (Skl MHI) [27] of human actions to implement HAR (human action recognition) that can work independently on the problem domain. Li et al. [28] treated the interactive actions as individual motions, combining global features and local features to identify human interactions. The algorithm framework for human interaction recognition is generally divided into whole body-based human interaction classification and individual-based human interaction classification [29]. Among them, the whole recognition method refers to describing the human interaction as a whole, including all the people involved in the interaction in the video. Guo et al. [30] categorized the multi-person interaction as individual layer or interaction layer, and proposed a hidden Markov model based on observation vector decomposition. Xiaofei Ji et al. [31] introduced a hierarchical structure of interactive action recognition based on the process of human interaction. According to chronological order, the actions are divided into action start time, action execution time, and action end time. Vahdat et al. [32] proposed an interactive action recognition algorithm based on chronological key poses by treating two interactions as two individuals, learning the model parameters of each individual, and then identifying them. However, this method could not capture the human interaction information. Such methods mainly deal with individual actions, which may interfere with the action classification results due to the existence of individual occlusion and self-occlusion.

This paper focused on human interaction recognition in video. In the existing video research algorithms for human interaction motion, many methods recognize the two sides of the action as a whole and lose the characteristic information that the individual brings. There are also a few algorithms that separate the two sides of the action into two separate individuals for recognition, but they ignore the characteristic information that the whole brings. Therefore, this paper proposed a human interaction recognition framework based on whole–individual detection. This method contains the characteristics of the whole information and the characteristics of the individual information. The whole video includes both sides of the action, and can extract global features such as the relative position and orientation of the action, and the individual detection video contains a single person, where the individual detailed action feature information can be extracted.

### 1.2. Contribution

First, we propose an improved fusion time-phase feature of the Gaussian model to obtain video keyframes and remove the influence of a large amount of redundant information. This method resolves the problem of the differences in action characteristics at different time periods.

Then, we propose a multi-feature fusion network algorithm based on parallel Inception and ResNet. This multi-feature fusion network not only reduces the network parameter quantity, but also improves the network performance; it alleviates the network degradation caused by the increase in network depth and obtains higher classification accuracy. This algorithm solves the problem of the complexity of human interaction feature extraction.

We have noted that the whole video contains the global features of both sides of action, and that the individual video contains the individual detail features of a single person. Therefore, we combined the whole video features with the individual video features, making full use of the feature information of the interactive video, which will help us solve the problem of the spatial complexity of human interaction.

## 2. Proposed Method

Due to the spatial complexity of human interaction, we propose a two-person interaction recognition algorithm based on whole–individual detection. The video includes two individuals of the action. Information can be extracted such as the relative position and orientation of the interactive action, and the individual detailed action features can be collected from the individual action video. As shown in Figure 1, in the motion video individual detection stage, the whole video is split into individual video (left) and individual video (right). This paper used HOG (histogram of oriented gradient) [33] and SVM (support vector machine) [34] and the Kalman tracking algorithm [35] to obtain the position of individuals for video detection as the two algorithms can reduce the impact of occlusion on pedestrian detection. In the feature extraction and model training stage, first, the image is preprocessed by data enhancement and normalization. Then, the video downsampling algorithm based on Gaussian distribution is used to improve the validity of the data. Finally, the parallel multi-feature fusion network is proposed for model training. In the action video recognition stage, the preliminary recognition results that are obtained by the whole video and the individual segmented video are fused. Then, we combine the preliminary recognition results decision level to obtain the final results.

### 2.1. Motion Video Individual Detection

Due to the spatial complexity of interaction features, the motion characteristics of individual movements and the characteristics of interaction movements, we proposed an interactive motion recognition framework based on whole–individual detection. In the individual detection part of the interactive action video, the moving target is detected for the action video. The video detection is performed according to the detection result, and the algorithm block diagram used is shown in Figure 2.

When the HOG feature descriptor and the pedestrian detection algorithm of the SVM classifier are applied to the interactive action video, loss detection and false detection phenomenon may take place. In this paper, a Kalman filter-based auxiliary tracking model was added to the pedestrian detection process to track each individual in the human interaction video. The trajectory and other information detected by consecutive frames provide reasonable prediction of the target’s next frame position. Human body loss detection is greatly reduced by using target tracking. As shown in Figure 3, a complete two-person interactive motion video is segmented into two individual motion videos, recorded as individual video (left) and individual video (right). First, we input the human interactive action video. Then, we extracted HOG features from the video. After that, we imported the parameters into the SVM classifier for training. Finally, we used the Kalman tracking algorithm to achieve individual detection.

### 2.2. Video Downsampling with Time-Phased Features

When using video images for feature extraction and training, the choice of video sampling methods will directly affect the generalization ability of the classification model. If there is no video downsampling, adjacent video will generate a lot of redundant information, which will increase the burden on the network. Considering the difference in the phase characteristics of the action time, this paper proposed a Gaussian model downsampling method that combines time-phase features. The Gaussian function is also called a normal distribution, and the expression of the probability density function is as shown in Equation (1).
(1)$$f(x)=\frac{1}{\sigma \sqrt{2\pi }}e^{-\frac{{\left(x-\mu \right)}^{2}}{2\sigma^{2}}}$$
where *μ* is the expected value of the Gaussian distribution and *σ* is the standard deviation of the Gaussian distribution, which is related to the magnitude of the Gaussian distribution. It can be seen from the distribution curve that the closer to the expected value, the higher the probability density value. There is a general rule in human interaction video: the closer to the trigger stage of the target action, the higher the distinction between actions. We associated this rule with a Gaussian probability density function that is very similar. In the public dataset taken in this paper, each action video sample could be divided into the action start period, action execution period, and action end period. Based on the analysis, it can be seen that the action execution period had significant feature differences, which can help to classify the movements, and the action frame was concentrated in the center. Analogous to the Gaussian model, we proposed a video downsampling method based on Gaussian probability distribution. In the process of video downsampling, we adopted different sampling methods according to the different action stages of the video. For the video from the beginning to the end of the target action, we sampled at small intervals, and for the redundant video outside the target action, we sampled at large intervals. The selection of the sampling interval needs to be obtained through repeated experiments. The video features will also be more differentiated after Gaussian downsampling processing.

### 2.3. Human Interaction Feature Extraction Based on Parallel Multi-Feature Network

In order to improve the accuracy of interactive action video recognition, a convolutional neural network based on parallel multi-feature fusion was proposed for the extraction of interactive feature information. The migration learning method is adopted in the feature extraction process [33]. A block diagram of the algorithm based on an Inception [36] and ResNet (Residual Network) [37] parallel multi-feature fusion network is shown in Figure 4.

The Inception V3 [38] network is an important breakthrough in the history of convolutional neural networks. We used the Inception pre-training network to obtain feature information. In general, in order to improve network performance, the most common way for researchers is to increase the depth and width of the network, but this approach will generate a huge number of parameters. As the number of network layers increases, the training process will be more cumbersome and will easily lead to over-fitting of the data. In order to ensure the performance of computing while expanding the network, the Inception network emerges. As shown in Figure 5, the network structure clusters the sparse matrix into more dense sub-matrices to improve the computing performance. The depth and width of the neural network are modified. The large convolution kernel is split into small convolution kernels of different sizes such as 1 × 1, 3 × 3, and 5 × 5.

The Inception structure uses different receptive fields to fuse different scale features, and the obtained image feature information is more complete. At the same time, the series of networks eliminates the full connection layer in the network output layer. This method greatly reduces the parameters of the entire network and improves the efficiency of the training process. With the rapid development of deep learning technology, the Inception network is constantly improving and innovating. For example, Inception V2 network [36] introduces the batch normalization layer, and uses two 3 × 3 convolution kernels to represent a 5 × 5 convolution kernel. The depth of the neural network improves its nonlinearity. The concept of asymmetric convolution is proposed in the Inception V3 network structure. In this structure, the large convolution kernel is further decomposed. In order to further reduce the number of parameters in the network and improve the speed of the operation, the network further decomposes the N × N scale convolution kernel and decomposes it into a 1 × N convolution kernel and an N × 1 convolution kernel. It can not only improve the operation speed, but also alleviate the data over-fitting phenomenon.

We continued to explore the feature extraction method based on residual neural network [37]. As the convolutional neural network goes deeper and deeper, a series of problems have emerged. In the process of network training, as the number of network layers deepens, gradient disappearance or gradient explosion may occur. These problems make the network difficult to converge. Researchers hope to increase the nonlinearity of neural networks by increasing the number of neural network layers. At the same time, they are trapped by the phenomenon of network degradation. Figure 6 shows the principle framework of the deep residual network.

In the residual network structure diagram of Figure 6, the input of x is directly transferred to the output as the initial result by ways of “shortcut connections”. The output result is H(x) = F(x) + x. When F(x) = 0, then H(x) = x. This is identity mapping (the output is equal to the input). Therefore, ResNet is equivalent to changing the learning goal, which is no longer to learn a complete output, but the difference between the target value H(x) and x, called residual F(x) = H(x) – x. Therefore, the following training goal is to approximate the residual result to 0, so that as the network deepens, the accuracy does not decrease.

There is a residual unit that can be expressed in the form of Equations (2) and (3):(2)$$y_{l}=h\left(x_{l}\right)+F\left(x_{l},W_{l}\right)\left(W_{l}=\left\{W_{l,k}|1\leq k\leq K\right\}\right)$$
(3)$$x_{l+1}=f(y_{l})$$
where *x_l_*, *y_l_* represent the input and output values of the *l_th_* neuron; *h(x_l_)* is the representative identity map; and *f(y_l_)* is the representation activation function. Under the condition of identity mapping: *h(x_l_) = x_l_*, *y_l_* = *f(y_l_)*, there is Equation (4).
(4)$$x_{l+1}=x_{l}+F(x_{l},W_{l})$$

Similarly, when it is coming to the *lth* layer, Equation (5) can be obtained.
(5)$$x_{L}=x_{l}+\sum_{i=l}^{L-1}F\left(x_{i},W_{i}\right)$$

It can be seen that when the number of layers is deeper and deeper, the output value of the network is related to the output of the residual in the previous layer and the output of the Lth layer is the sum of the output values of the residuals of the previous layer. In back-propagation, calculation of the partial derivative of the loss function *ε* is as Equation (6).
(6)$$\frac{\partial \varepsilon }{\partial x_{l}}=\frac{\partial \varepsilon }{\partial x_{L}}\frac{\partial x_{L}}{\partial x_{l}}=\frac{\partial \varepsilon }{\partial x_{L}}\left(1+\frac{\partial }{\partial x_{l}}\sum_{i=l}^{L-1}F\left(x_{i},W_{i}\right)\right)$$

It can be seen from the equation that the process of gradient derivation avoids the possibility that the value in the multiply state is 0, thus avoids the trouble caused by the disappearance of the gradient. After the network is improved in this way, even if the network is deeper, the results of network training will not be too deviated. This improved method avoids the problems of gradient disappearance and gradient explosion, which are caused by the increase in the number of network layers, which leads to maintaining a good network performance.

Taking the interactive video in the UT dataset [39] as an example, we used the Inception V3 pre-training model based on the ImageNet dataset. According to the requirements of the model, the image in the activity video was adjusted to a size of 299 × 299. The output of the last layer of the average pooling layer was used as the result of the preliminary feature information extraction of the Inception network. The results are stored as a series of feature files, each of which generates a feature file. Each segment of the video generates a feature file. During the experiment, each group of action videos was cut into 40 frames, so the output size of the network was 2048 × 40. During the experiment, for the ResNet pre-training network, the image was cropped to a uniform 224 × 224 size at the feature extraction stage due to the input data requirements of the pre-training model, Similarly, each video will generate a feature file, which is extracted based on the residual neural network with a feature size of 2048 × 40.

The proposed network is a multi-feature fusion convolutional neural network based on Inception and ResNet. At the feature extraction level, feature extraction is performed using the Inception V3 network and ResNet, respectively. Then, the two extracted features are fused at the fully connected layer. Then, training is continued. After multi-feature network convergence, the image features will be more abundant, which is conducive to improve the accuracy of human interaction video classification and recognition. According to the name of the video, this paper fused two feature files generated by each video file to form a feature file with a size of 4096 × 40 for later training and classification.

### 2.4. Whole-Individual Detection Based on Decision-Level Fusion

The decision-level fusion uses different features to obtain the classification results, and then the experimental results are merged. In the classification recognition phase, the whole video, individual video (left), and individual video (right) all produce a preliminary classification result based on probability. In order to make better use of the feature information of video images and improve the action recognition accuracy of interactive video, from the perspective of probability fusion, this paper fused the preliminary classification results at the decision level to obtain the final classification results. As shown in Equation (7), probabilistic weighting is used to fuse the three classification results of each set of action videos to obtain the final classification result:(7)$$R_{Final}=R_{Overall}\times P_{Overall}+R_{Left}\times P_{Left}+R_{Right}\times P_{Right}$$

Among them, *R_Final_* is the final recognition result, *R_whole_* is the whole video classification result of the double, *R_Left_* represents the classification result after training using the individual video (left), and *R_Right_* represents the classification result obtained by using the individual video (right) for model training. *P_whole_*, *P_Left_*, *P_Right_* are the weighted probability of the whole video classification result of the two person, the weighted probability of the individual video (left) classification result, and the weighted probability of the individual video (right) classification result, respectively. The weighted probability value was obtained by comparing and analyzing the repeated experiments. A human body interactive video recognition block diagram based on decision level fusion is shown in Figure 7. As shown in Figure 7, after the video sequence is processed by a parallel multi-feature network, we can obtain the global and individual features. After the global feature and individual features are classified by Softmax, we can obtain a preliminary classification result based on probability. In the final classification stage, in order to make better use of the feature information of video images and improve the recognition accuracy of interactive video, this paper combined the preliminary classification results from the perspective of probability fusion to obtain the final classification results.

## 3. Results

### 3.1. Experimental Platform and Experimental Data

This paper conducted the experiments on a computer with a NVIDIA GTX 1080Ti graphics card. The experiment used Tensorflow for the deep learning network framework, Python for programming, MATLAB for other data analysis and preprocessing stages; the software used were Pycharm2018 and MATLAB2016.

This paper selected the UT interaction dataset. The UT dataset is a common human interaction video dataset. In order to ensure the effectiveness of the algorithm, the original data are generally divided into a training set and a verification set. This paper used K-fold cross validation (K-CV). This method divides the original data into K groups on average (K is generally greater than or equal to 2, the actual operation generally starts from 3, we only try to take 2 when the amount of raw data is small. Therefore, in this paper, we selected K = 3 for the experiment.) Each subset of data is used as a verification set, and the remaining K-1 subsets of data are used as training sets. In this way, K models are obtained. Finally, we used the average number of classification accuracy of the final validation set of these K models as the final result. This method is the most widely used as K-CV can effectively avoid the occurrence of over-fitting and under-fitting and the final result has high reliability. The UT-Interaction dataset contains videos of continuous executions of six classes of human–human interactions: shake-hands, point, hug, push, kick, and punch. Ground truth labels for these interactions are provided including time intervals and bounding boxes. There is a total of 20 video sequences whose lengths are around one minute. Each video contains at least one execution per interaction, providing us with eight executions of human activities per video on average. Several participants with more than 15 different clothing conditions appear in the videos. The videos are taken with the resolution of 720 × 480, 30 fps, and the height of a person in the video is about 200 pixels. We divided videos into two sets. Set 1 was composed of 10 video sequences taken on a parking lot. The videos of set 1 were taken with slightly different zoom rate, and their backgrounds were mostly static with little camera jitter. Set 2 (i.e., the other 10 sequences) were taken on a lawn on a windy day where the background is moving slightly (e.g., tree moves), and they contain more camera jitters. From sequences 1 to 4 and from 11 to 13, only two interacting people appear in the scene. From sequences 5 to 8 and from 14 to 17, both interacting people and pedestrians are present in the scene. In sets 9, 10, 18, 19, and 20, several pairs of interacting people execute the activities simultaneously. Each set has a different background, scale, and illumination. As shown in Figure 8 and Figure 9, UT set 1 is an example of an action in the background of a parking lot, and UT set 2 was photographed on a windy lawn. For the entire video dataset, the video capture had different backgrounds, different resolutions, and different lighting conditions, which all bring challenges to the experiment of human interaction recognition.

In addition, in order to verify the applicability of this algorithm, we selected the interactive action video in UCF101 [40] for extended experiments. Since we are interested in human interactive videos, we selected a part of the interactive videos in the UCF101 dataset for training and testing. Our experiment on the UCF101 dataset was only to verify the effectiveness and versatility of the method. At present, the experiments of interactive action recognition are usually compared on the UT dataset. UCF101 is an action recognition dataset of realistic action videos collected from YouTube, having 101 action categories. This dataset is an extension of the UCF50 dataset, which has 50 action categories. With 13,320 videos from 101 action categories, UCF101 had the largest diversity in terms of actions and contained the presence of large variations in camera motion, object appearance and pose, object scale, viewpoint, cluttered background, illumination conditions, etc., which made it the most challenging dataset to date. As most of the available action recognition datasets are not realistic and are staged by actors, UCF101 aims to encourage further research into action recognition by learning and exploring new realistic action categories. The videos in 101 action categories were grouped into 25 groups, where each group can consist of 4–7 videos of an action. The videos from the same group may share some common features such as similar background, similar viewpoint, etc. We randomly divided the original data of the UCF101 interactive video into two groups: one as the training set and one as the verification set. We then used the training set to train the classifier, and then used the verification set to verify the model, and recorded the final classification accuracy as the result.

More experimental parameters are as follows. We chose the Adam optimizer in the experiment, the learning rate = 10^−5^, and decay = 10^−6^. The loss function uses categorical cross entropy. During training, batch size = 32. During the training process, the image in the action video was adjusted to an image of size of 299 × 299. In the feature extraction stage, the image was cropped to a uniform size of 224 × 224. During the experiment, each group of action video clips was 40 frames, so the output size of the network was 2048 × 40.

### 3.2. Analysis of Results

In order to verify the improved effect of video acquisition based on Gaussian downsampling, we took the interactive video of UT dataset 1 as an example. We adopted equal interval downsampling video processing and Gaussian model based downsampling video acquisition to perform UT whole video preprocessing. In this paper, we selected the Inception network and ResNet for feature extraction and classification recognition processing, and the experimental results are shown in Table 1.

It can be seen from Table 1 that the Gaussian model based downsampling method could improve the accuracy of human interaction recognition. The experimental process found that the improved fusion time-phased Gaussian model downsampling algorithm had an improved effect on the recognition of punching and pushing human interaction. However, the recognition and improvement of other actions were not obvious. This might be due to the limiting rules (some action video durations were shorter) of the video in the chosen dataset. Furthermore, there was no big difference in the choice of sampling methods.

In order to verify the human–interaction recognition algorithm based on the whole–individual detection proposed in this paper, a comparative experiment was carried out on the UT dataset. The UT interactive dataset generates individual video (left) and individual video (right) after the previous interactive video detection process. In this paper, UT individual video (left), UT individual video (right), and whole video were used for the experiments. In order to obtain reliable experimental classification and comparison, the experimental process was based on an Inception and ResNet parallel multi-feature fusion network algorithm. In the video downsampling process stage, we used a Gaussian model with improved fusion time phase features.

The experimental results of the UT dataset 1 were analyzed. Among them, the preliminary classification results of the individual actions obtained by the individual (left) video are shown in Figure 10, and the preliminary classification results obtained by the individual video (right) are shown in Figure 11. The preliminary classification results based on the whole video are shown in Figure 12. Finally, we combined the whole video and preliminary classification results of two groups of segmented video components for decision level fusion. The classification results of each action video after fusion are shown in Figure 13. In our experiments, the vertical axis of the confusion matrix is the actual actions, and the horizontal axis is the recognized actions.

The results of our experiments on the UT human interaction dataset are shown in Figure 14 and Figure 15. It can be seen from Figure 14 that the accuracy of the training set and the validation set steadily increased, while Figure 15 is an image of the loss function. In terms of a specific sample, the loss function refers to the gap between the value predicted by the model and the true value. For a sample (x_i_, y_i_), y_i_ is the true value and f(x_i_) is our predicted value. Use the loss function L(f(x_i_), y_i_) to represent the gap between the true value and the predicted value. At the same time, it can be seen from Figure 15 that the loss function of the training set and the validation set both steadily decreased. In the end, they tended to coincide.

After the deciding level fusion, the whole accuracy of the video classification results is shown in Table 2. In the UT set 1 human interaction dataset, the individual video (left) obtained 83.3% recognition accuracy, the individual video (right) accuracy rate reached 75%, and the recognition accuracy based on UT whole video reached 86.1%. In contrast, the recognition result of the UT dataset was better than the video classification result after the individual split. The decline in recognition accuracy after individual detection may be due to the easy loss of feature information of interactive actions during video detection. In individual-split video, kicking, punching, and pushing actions always have a side action performer in an evasive state, thus the motion discrimination degree is small. Thus, the recognition accuracy of the individual split video is relatively low.

The comparison experiment results are shown in Table 2. For the selection of the optimal weights, we adopted the method of weight traversal. After the decision level fusion, the recognition accuracy of interactive video was significantly improved. From the analysis of the recognition accuracy of each action, it can be seen that the accuracy of the classification results of pointing and pushing was higher. This was not hard to predict since the characteristics of these two actions were more obvious. The classification result of the punching action needs to be improved because it is easily recognized as a push action and a handshake action. The kicking action was easily confused with the pushing action due to the low discrimination of degree of motion. The experimental results showed that when the detection video and the corresponding original whole video classification results were fused according to a certain probability, the classification accuracy of the handshake action and the push action was significantly improved in the human interaction. In addition, in order to prove the credibility of this method, we also performed extended experiments on the UCF101 dataset. In the UCF101 human interaction dataset, the individual video (left) obtained a recognition accuracy of 75.6%, the individual video (right) obtained a recognition accuracy of 76.8%, and the UCF101 whole interactive video obtained a recognition accuracy of 81.8%. Thus, it can be seen that the method of individual detection and whole fusion proposed in this paper can improve the accuracy of human interaction recognition.

The experimental results of this paper were compared with the classification results of other experimental methods on the UT dataset in recent years, and the experimental results obtained are shown in Table 3. Huang et al. [30] used the HIS color space model to analyze the characteristics of the direction gradient histogram for different channels. Then, multi-channel fusion yielded 81.7% recognition accuracy. Mahmood et al. [41] proposed a new human interaction recognition (HIR) method that analyzes from local features, captures intensity changes, and distance from point to point. Time-based relationships identify key body points throughout the body contour and they used this method to extract the spatiotemporal characteristics of each different interaction. In their paper, the recognition in the UT set 1, two dataset experiments obtained an accuracy of 83.5% and 72.5%, respectively. Kong et al. [42] introduced an interactive phrase descriptor to represent the human interaction movement relationship and obtained a recognition rate of 88.33% in the UT dataset experiment. Shariat et al. [43] proposed a detection alignment model to improve the similarity measure between different action sequences, where they obtained a recognition rate of 91.57%. Guo et al. [44] proposed a new local descriptor based on the traditional descriptor LBP, and extended it to a space–time space. They obtained a recognition accuracy of 91.42% by using the neighborhood information of the three-dimensional cube. Using the method proposed in this paper, the experiment was carried out with the UT dataset, and we finally obtained a recognition rate of 91.70%. In order to prove the effectiveness of this method, we performed verification experiments on the UCF101 dataset and obtained a recognition rate of 85.43%. Therefore, it can be seen that the human-interaction recognition algorithm based on the whole–individual detection proposed in this paper can improve the accuracy of human interaction recognition rate.

## 4. Conclusions

In this paper, the human-interaction recognition algorithm based on whole–individual detection was proposed. The experimental verification and analysis work was carried out with the human interaction UT dataset. In the stage of feature information extraction of human interaction, we proposed human-interaction recognition in a parallel multi-feature fusion network. Compared with a single feature information extraction network, the fusion network improved the whole recognition accuracy of interactions. Regarding the complexity of interactive action features, we proposed an improved human body interaction recognition method based on the whole–individual detection. For the differences in action characteristics at different time periods, a Gaussian-based video downsampling method was proposed. This method makes the data acquisition more consistent with the characteristics of each action time. The results show that the whole–individual detection human interaction recognition method based on decision-level fusion proposed in this paper can improve the accuracy of classification recognition. In this paper, the complexity of the algorithm was relatively high. Subsequent work will further improve the complexity of the algorithm to improve the efficiency of recognition.

## Figures and Tables

**Figure 1 sensors-20-02346-f001:**
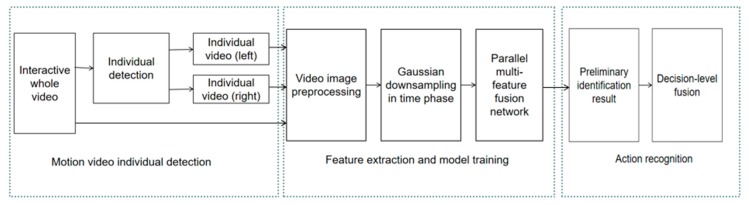
Human interaction recognition based on whole–individual detection.

**Figure 2 sensors-20-02346-f002:**
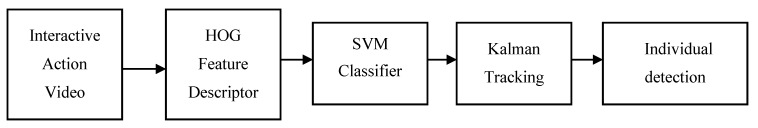
Block diagram of the individual detection algorithm for interactive video.

**Figure 3 sensors-20-02346-f003:**
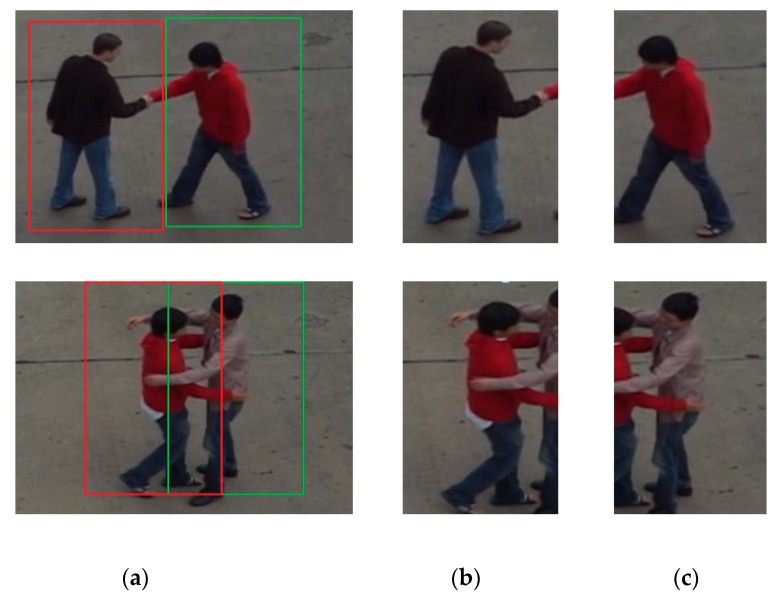
Video individual detection diagram: (**a**) Original video; (**b**) Individual video (left); (**c**) Individual video (right).

**Figure 4 sensors-20-02346-f004:**
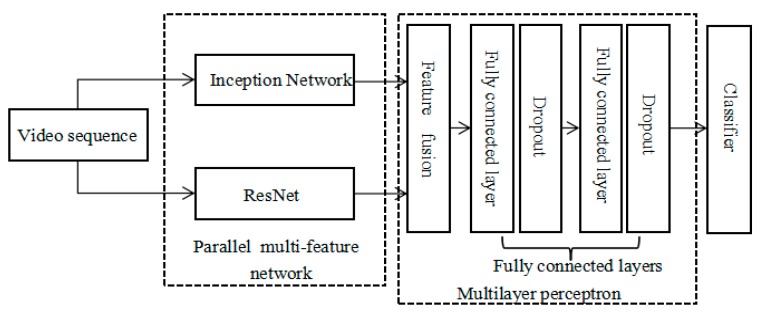
Block diagram of the parallel multi-feature fusion network algorithm.

**Figure 5 sensors-20-02346-f005:**
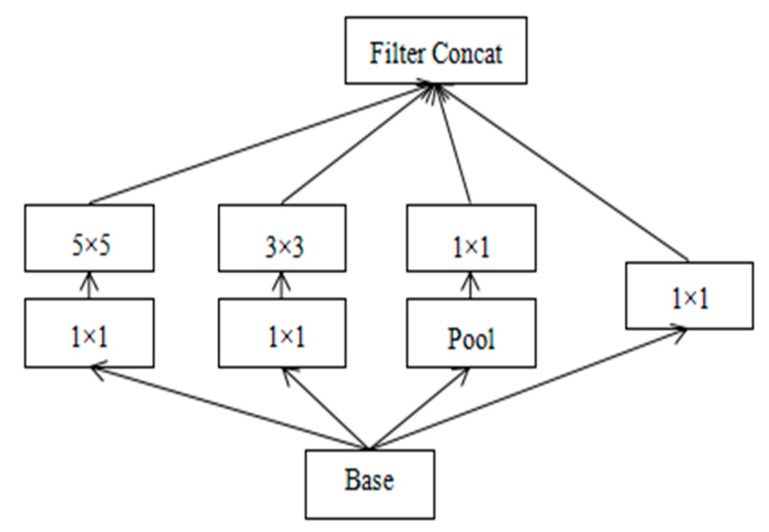
Inception module.

**Figure 6 sensors-20-02346-f006:**
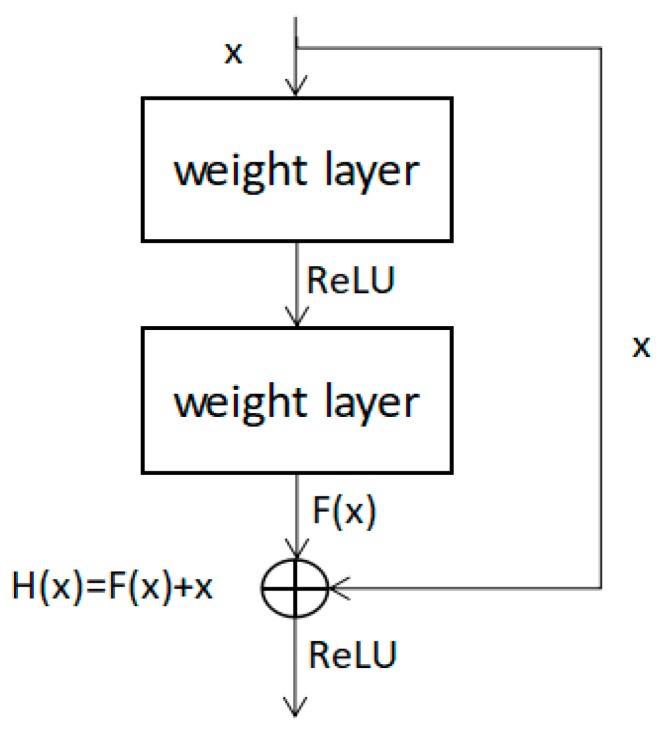
ResNet basic principle structure diagram.

**Figure 7 sensors-20-02346-f007:**
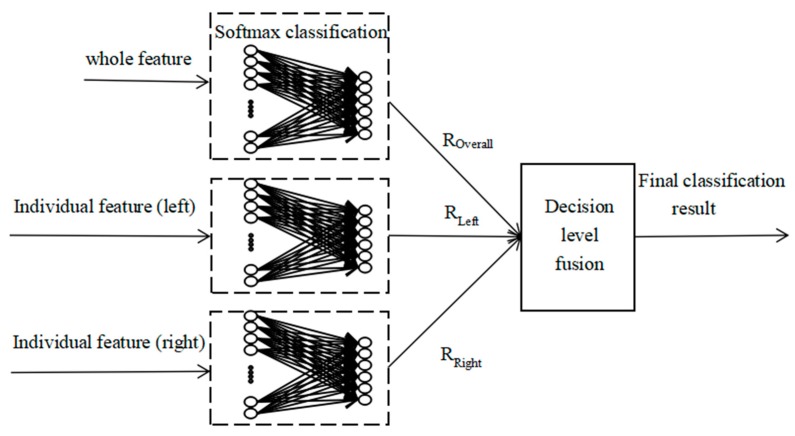
Human body interactive video recognition based on decision level fusion.

**Figure 8 sensors-20-02346-f008:**
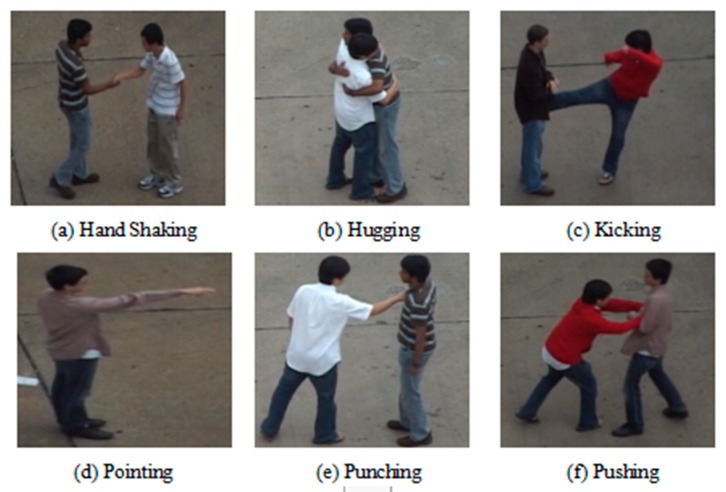
Example of UT set 1 dataset.

**Figure 9 sensors-20-02346-f009:**
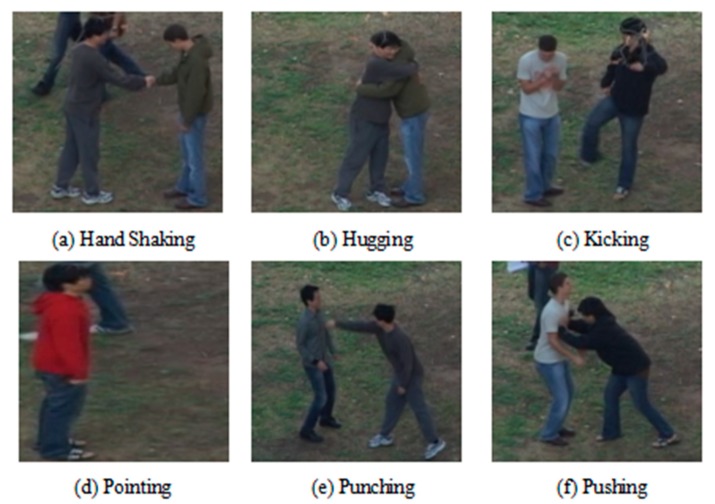
Example of the UT set 2 dataset.

**Figure 10 sensors-20-02346-f010:**
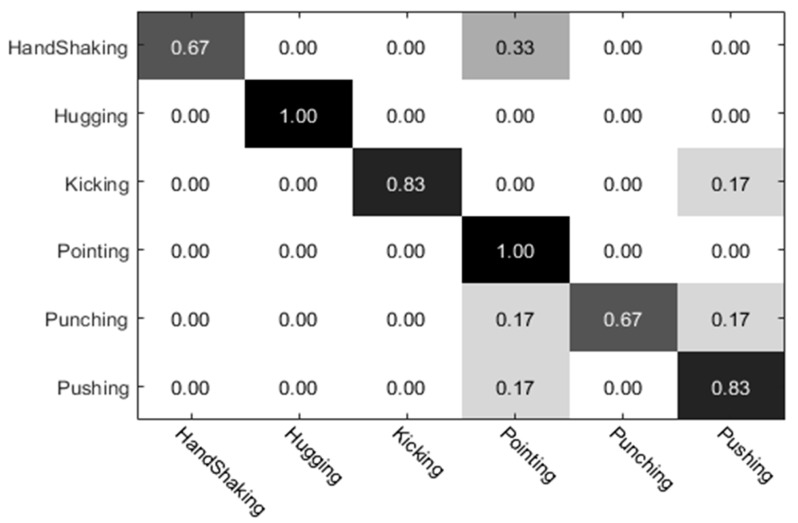
Individual video (left) classification results.

**Figure 11 sensors-20-02346-f011:**
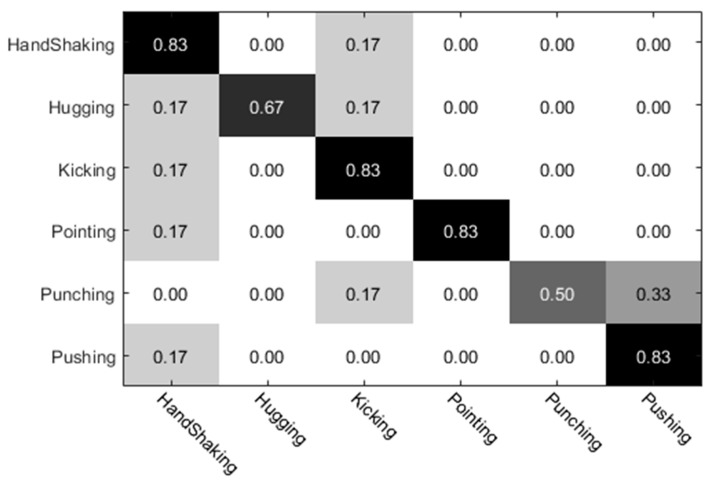
Individual video (right) classification results.

**Figure 12 sensors-20-02346-f012:**
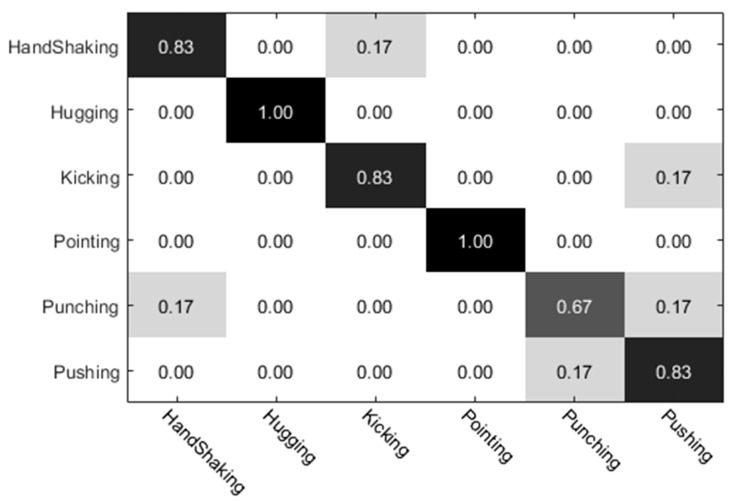
Whole video classification results.

**Figure 13 sensors-20-02346-f013:**
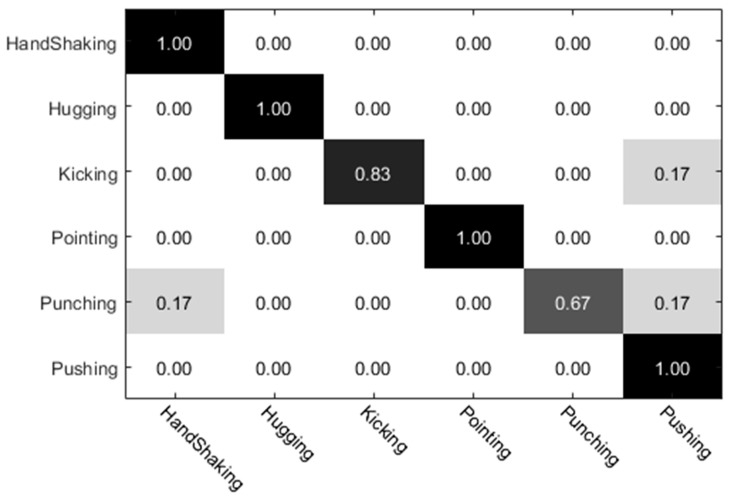
Classification results of human interaction based on whole–individual detection.

**Figure 14 sensors-20-02346-f014:**
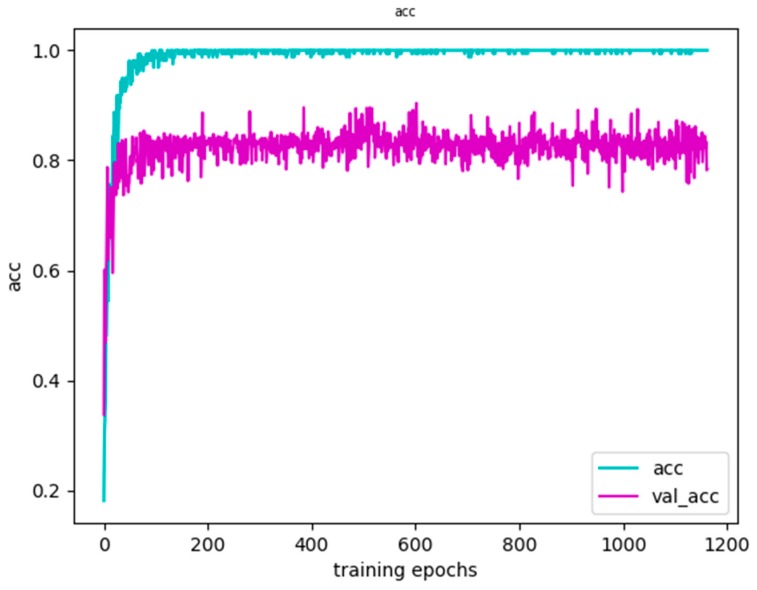
Accuracy of the training results.

**Figure 15 sensors-20-02346-f015:**
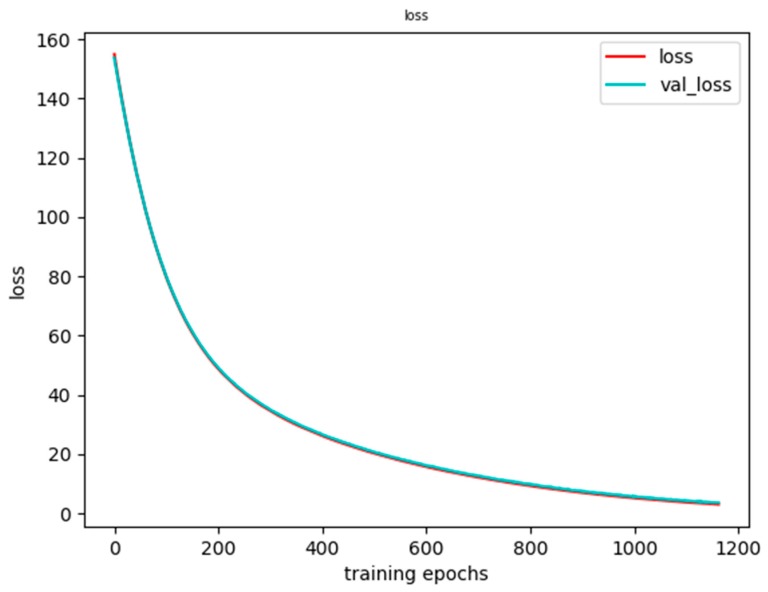
Loss function of the training results.

**Table 1 sensors-20-02346-t001:** Comparison of recognition accuracy of different sampling methods.

Sampling Method	Inception (%)	ResNet (%)	Multi-Feature Fusion (%)
Equal interval sampling	66.7	72.2	83.3
Downsampling based on Gaussian model	69.4	77.8	86.1
Equal interval sampling	66.7	72.2	83.3

**Table 2 sensors-20-02346-t002:** Comparison of the individual video and whole video recognition accuracy.

Recognition Methods	UT Set 1 Recognition Accuracy (%)	UT Set 2 Recognition Accuracy (%)	UCF101 Interactive Recognition Accuracy (%)
Individual (left)	83.3	77.8	75.6
Individual (right)	75.0	72.2	76.8
whole	86.1	83.3	81.8
Fusion of this paper	91.7	86.1	85.4

**Table 3 sensors-20-02346-t003:** Comparison of the recognition accuracy of different identification methods.

Recognition Methods	UT Set 1 Recognition Accuracy (%)	UT Set 2 Recognition Accuracy (%)	UCF101 Interactive Recognition Accuracy (%)
HIS color space model [30]	81.70	--	--
Interactive phrases [42]	88.33	--	--
detection alignment model [43]	91.57	--	--
Novel 3D gradient LBP descriptor [44]	91.42	--	--
Method of this paper	91.70	86.10	85.43

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
