# Peer review of "Human Interaction Recognition Based on Whole-Individual Detection"

_sensors, 2020, doi:10.3390/s20082346_

Round 1

Reviewer 1 Report

In this manuscript, the authors proposed a whole-individual segmentation method for interactive action recognition. The primary problem of the manuscript is that its presentation is a little poor, which makes it difficult to understand the work. To satisfy the requirement of publication, further improvements are necessary. Detailed comments are listed below.

  1. Though the authors give a detailed review of video-based algorithms for human motion recognition, a review of algorithms for human interaction motion is lacked, which is actually the main focus of this paper. In addition, in the last paragraph of Section 1, the authors should give a better explanations of "whole information" and "characteristics of the individual information", which are not clear enough. I suggest the authors to give an example to better illustrate these two conceptions.
  2. Line 87-88, what is HOG short for? The authors should give the full name of abbreviations when they first appear. This comment is also valid for other abbreviations. In addition, please provide references for HOG and SVM pedestrian detection algorithm and Kalman tracking algorithm.
  3. More details about HOG feature descriptor and the pedestrian detection algorithm of the SVM classifier should be provided in the manuscript. The authors should explain how they work in the block diagram shown in Figure 2.
  4. Line 134, what do the authors exactly mean by "video execution stage" and “the degree of discrimination”? The presentation is ambiguous.
  5. Line 139-142, “During the video … gets bigger”, this sentence is difficult to understand. Please rephrase it. It is still unclear to me how the downsampling works. Is it used to reduce the length of the video are only select the interval that the authors are most interested in?
  6. Line 148, please provide references for Inception and ResNet networks.
  7. In Figure 4, there are two operations of "Fully connected layer and Dropout" in the block diagram. What is the difference between them? The authors should give more information in the figure to avoid misunderstanding.
  8. In Figure 5, what is the difference between the path "Base->Pool->1*1->Filter Concat" and "Base->1*1->Filter Concat"?
  9. Line 169, a reference should be given for Inception V2 network.
  10. Clear illustration of Figure 6 is required.
  11. Line 240, variable names are overlapped with texts.
  12. The authors should give a clearer explanation of how the training and verification were performed. For example, which data are used for training and which data for verification. In addition, the authors mentioned several datasets in the manuscript, such as UT dataset and UCF101 dataset. For UT dataset, it includes UT set 1 and UT set 2. It is unclear how these datasets were used in data analysis, which makes it difficult to understand the reported results. The authors wrote “In order to prove the applicability of the method in this paper, experimental verification is performed on the UCF101 dataset”. Did the authors mean the UT dataset was used for training while the UCF101 dataset is used for verification? I also noticed that interactive actions observed in UCF101 datasets are not the same with those in UT dataset. I am wondering how the authors verify the algorithm when these actions were not included in the training dataset.
  13. Are UT dataset and UCF101 dataset public resources? If so, please provide references/links to them.
  14. Since a more complex algorithm usually produces better performance but more computation complexity, I suggest the authors to also compare computation duration apart from recognition accuracy.
  15. In the confusion matrices shown in Figure 10-13, please explain the actual and recognized actions are displayed in column or in row.
  16. What is the definition of loss rate? Why its value can be larger than 1 in Figure 15?

Some sentences are difficult to follow or have grammar errors. Please rephrase them. Some examples are:

Line 45-46, “Silambarasi et al. [24] based on … 3D space-time planes.” Please rephrase the sentence.

Line 46, “They use time-view …” should be revised as “They used time-view …”.

Line 90, “Firstly” should be revised as “firstly”.

Line 105, “is as shown” should be revises as “is shown”.

Reviewer 2 Report

Authors proposed fusion base Human interaction recognition algorithm in the paper,  I have few concerns:

(A) As authors used deep learning models ResNet and Inception for the fusion of the features, the reader can be confused with "Segmentation" word, as they are not using box based detection, Authors need to use the appropriate word in the title and the whole text (for example "ROI" or "Detection").

(B) Considering the Abstract of the paper, I feel the abstract weak in terms of "problem definition/ problem introduction/ Research need"==> "How existing methods are lacking ?" ==> "Solution"==> "Proposed method".

(C) if possible provide the full name of the UT dataset once in the abstract

(D) The introduction is the layout that describes the need for the research, please separate it from the "Related Works", Make separate sections for "Introduction" and "related works". With quality references, the authors need to create the whole story to prove the importance of interaction/ action recognition (in the introduction section). 

(E) Based on the key points of your research design (Used in the abstract "What is different in your research that others don't have"), please add a separate section "Contribution", in which the paper contributions should be mentioned briefly in bullets.

(F) Coming towards the research design, the authors didn't mention in the paper why they used ResNet and Inception ??? is there any specific reason? considering the Human interaction recognition the method should be real-time or near to it. Why they do not use the MobileNet, SqueezNet? etc. 

(G) I am a little confused about the research contribution of the paper, the methods that are used are already developed (ResNet, Inception). The author need to strongly prove their research contributions.

(H) In my opinion, the sectioning of the paper is required (1) introduction (2) related work (3) contribution (4) proposed method (5) Results (6) discussion (7) Conclusion

(I) UT dataset is own created dataset? if not where is the reference? please provide proper reference and details about the dataset, annotations (who designed it ?), number of training/ testing/ validation images/ dataset split details?

(J) why the authors hide the implementation details?, How can I believe the results when the complete environment is not described? "Learning rate", "optimizer details" "learning rate schedule", "batch size", "image size" etc.

(K) On page 13 line 354 Huang et al. [18], the reference [18] is not Huang et al. 

(L)  provide the reference of UCF101 and details

(M) provide the details of the experiment of UCF101 dataset, authors just "tested" on UCF101 or "trained and tested" on UCF 101 dataset, if there are other studies for UCF101 dataset please compare your method with those in a separate table 

Round 2

Reviewer 1 Report

The authors have addressed most of my concerns. However, there are still some points misunderstood or not well addressed.

  1. For point 10 I raised in the previous review report, I wanted the authors to explain the block diagram shown in Figure 6, rather than adjusting the size of the figure.
  2. If I understand correctly, the authors perform training and testing using the same dataset, right? If so, the verification results are not convincing enough, because it is unknown how the trained algorithm works with new dataset (not included in the training dataset). I suggest the authors to verify the algorithm with cross-validation.
  3. For point 15 and 16 I raised in the previous review report, though the authors have answered my questions, I didn’t find these explanations in the manuscript. I think it is necessary to add these explanations in the manuscript as well.

Reviewer 2 Report

Overall authors addressed most of the questions asked in the previous revision. but need to fix these things prior to publication:

1) Do not mention the reference of UT-dataset in the abstract,  just provide the reference in the main body of the paper.

2) The answers that are provided in the "Author response" are detailed enough. But I don't feel any worse for putting these answers into the manuscript briefly.  

3) put dataset details and information about the number of training and testing video/image in the paper (point I and L) 

4) The paper length is not the excuse for not putting the implementation details (response for comment J) if these details are not available how a reader can reproduce the experiment and results?

5) Put the details of the experiment pointed out in comment "M" and reflect it in the main text of the paper
